# Heartburn-Related Internet Searches and Trends of Interest across Six Western Countries: A Four-Year Retrospective Analysis Using Google Ads Keyword Planner

**DOI:** 10.3390/ijerph16234591

**Published:** 2019-11-20

**Authors:** Mikołaj Kamiński, Igor Łoniewski, Agata Misera, Wojciech Marlicz

**Affiliations:** 1Sanprobi Sp.z.o.o. Sp.K., 70-535 Szczecin, Poland; 2Faculty of Medicine I, Poznan University of Medical Sciences, 60-780 Poznan, Poland; 3Department of Biochemistry and Human Nutrition, Pomeranian Medical University, 70-204 Szczecin, Poland; sanprobi@sanprobi.pl; 4Department of Child and Adolescent Psychiatry, Charité Universitätsmedizin, 13353 Berlin, Germany; agatamisera@gmx.de; 5Department of Gastroenterology, Pomeranian Medical University, 70-204 Szczecin, Poland; marlicz@hotmail.com

**Keywords:** Google, AdWords, heartburn, epidemiology, prevalence, United States, trends, Germany, gastroesophageal reflux disease, infodemiology

## Abstract

The internet is becoming the main source of health-related information. We aimed to investigate data regarding heartburn-related searches made by Google users from Australia, Canada, Germany, Poland, the United Kingdom, and the United States. We retrospectively analyzed data from Google Ads Keywords Planner. We extracted search volumes of keywords associated with “heartburn” for June 2015 to May 2019. The data were generated in the respective primary language. The number of searches per 1000 Google-user years was as follows: 177.4 (Australia), 178.1 (Canada), 123.8 (Germany), 199.7 (Poland), 152.5 (United Kingdom), and 194.5 (United States). The users were particularly interested in treatment (19.0 to 41.3%), diet (4.8 to 10.7%), symptoms (2.6 to 13.1%), and causes (3.7 to 10.0%). In all countries except Germany, the number of heartburn-related queries significantly increased over the analyzed period. For Canada, Germany, Poland, and the United Kingdom, query numbers were significantly lowest in summer; there was no significant seasonal trend for Australia and the United States. The number of heartburn-related queries has increased over the past four years, and a seasonal pattern may exist in certain regions. The trends in heartburn-related searches may reflect the scale of the complaint, and should be verified through future epidemiological studies.

## 1. Introduction

Heartburn can occur incidentally or may be a chronic symptom of gastroesophageal reflux disease (GERD) [1]. This complaint requires differential diagnosis in order to exclude cancer and other non-malignant conditions, such as eosinophilic esophagitis, functional dyspepsia, hypersensitive esophagus, and gastroparesis [2,3]. In adults, the global burden of GERD-related symptoms ranges from 6 to 30% per week [3,4]. Further, Dent et al. estimated that, in the Western world, the incidence of heartburn and/or acid regurgitation at least once a week is approximately five per 1000 person-years [5]. Another notable development is that the use of antacids and proton pump inhibitors (PPIs), which are treatment methods for heartburn, is increasing [6], and such approaches can negatively impact human health e.g., increasing odds of *Clostridum difficile* infection [7,8,9]. The continuing increase in the global prevalence of GERD-related symptoms may be caused by increases in the number of people with obesity [3], the aging of populations [4], and a surge in the consumption of nonsteroidal anti-inflammatory drugs and/or aspirin [4].

Limited access to public health-care has caused many individuals to perceive the Internet as an attractive alternative to real-life health-care professionals [10,11]. Further, the internet provides the convenience of immediate access to resources associated with the condition in question, as well as the comfort of potential support from other users and anonymity [12]. Previous research has estimated that 65–90% of internet users search for health-related information online [13], and that approximately 80% of these users trust the information they find [12]. Thus, it is probable that internet traffic reflects the real-world health problems of populations [14].

The approach of investigating data from an electronic medium for epidemiological studies is called infodemiology (information epidemiology) [15]. Infodemiology has the potential to search for trends that are unavailable in classic epidemiological studies. For instance, people may experience moderate ailment and search for relief on the Internet which may be explained by a reluctance to seek help for minor ailments in a healthcare professional office. For this reason, analysis of the Internet data may reveal poorly investigated relationships. 

Google is the most popular search engine among Internet users in Western countries [16]. The “Keywords Planner” function of Google Ads, which is linked to Google’s search engine, is a specially designed marketing tool for e-commerce campaigns, and has also been used as a recruitment engine for clinical trials [17,18], and survey study [19]. Moreover, Google Ads may also be a useful infodemiological tool, as it can deliver estimated search volumes for keywords associated with given terms for a chosen language, region, and timeframe [20]. These properties were intensively used in a series of dermatological studies [20,21,22,23,24]. In comparison to Google Trends, the Keyword Planner presents a range of searches per month of each search term. Google Trends present search volume as an index from 0 to 100 [14]. Considering the high prevalence of heartburn in the Western world, we can assume that the analysis of Google data will provide a unique insight into the scale of this complaint. This may reveal under-researched relationships that could be verified through further real-world epidemiological studies.

Considering the above, in the present study, we aimed to retrospectively investigate frequency, yearly and seasonal trends of heartburn-related searches made by users from Australia (AU), Canada (CA), Germany (DE), Poland (PL), the United Kingdom (UK), and the United States (US).

## 2. Materials and Methods

### 2.1. Search Strategy for Data Extraction

For this infodemiological study, we extracted research volumes for the term “heartburn” (German: das Sodbrennen, Polish: zgaga) and generated keywords for the period of June 2015 to May 2019 using the Google Ads Keywords Planner (ads.google.com/aw/keywordplanner). The tool allows to analyze search volumes of terms from the last 48 months. The data were generated separately for each region (chosen language): AU (English), CA (English), DE (German), PL (Polish), UK (English), US (English); search network: Google. The considered regions were countries. A screenshot of the search engine keywords is presented in Appendix A. The data was collected during 08–14 July 2019. The initial search terms were types without quotation marks. We collected the data for each month separately. The search network was set to “Google”, and we showed all proposed keywords. The planner generated a list of proposed terms associated with “heartburn” and expressed the search volume range for each keyword as an exponent of 10, e.g., 100–1000. Among the generated keywords are those with spelling mistakes. Two of three authors (M.K., I.Ł., A.M.) independently analyzed the generated list of keywords and create the final lists. Any inconsistencies in the lists were referred by the senior author (W.M.). Since the Keywords Planner propose keywords that, while related to the term “heartburn,” were not associated with abdominal pain, we excluded all keywords not related to the target complaint (e.g. “indigestion,” “nora ephron heartburn,” “alicia keys heartburn”). 

Additionally, we distinguished keywords that are related to questions regarding heartburn (e.g., “what is heartburn?”); its causes, symptoms, and treatment (including home-based methods and herbal remedies); relation to diet; and patient groups. Keyword “categories” (based on the character of each keyword) were created. A keyword (correct and incorrect orthographical variations) can be attributed to multiple categories (e.g., the keyword “heartburn treatment pregnancy” could be categorized as “treatment” or “pregnancy”), and a keyword category may also comprise multiple keywords (e.g., the category “treatment” contains “heartburn medicine” and “severe acid reflux remedies”). In this manuscript, we only present categories of keywords that represent at least 1% of the total number of heartburn-related queries in each country. An illustration of the data collection and manipulation process is presented in Appendix A.

### 2.2. Statistical Analysis

For all countries, we calculated the arithmetical mean of the sum of monthly search volume for each keyword category and all keywords. The data was presented as the total number of queries during the analyzed period for each keyword category and all keywords. Additionally, we divided the total search volume of each category by the total number of searches in the analyzed period, and separately by the number of Google users in each country, as declared by the Keywords Planner. Additionally, we presented the mean Google search engine market share in investigated countries in the years 2015–2019 [16]. Specifically, the ratio was expressed as “the number of queries”: “1000 Google-user years.” Moreover, we presented the number of searches for specific keyword categories as a percentage of the total number of searches associated with “heartburn” in each analyzed region. The number of queries made during each season and year were presented as medians (interquartile range). To compare the search volumes of each season and year, the Kruskal–Wallis test, with a post-hoc pair-wise Mann-Whitney U test, was performed. *p* values of < 0.05 were considered to indicate significant difference. The considered seasons were: spring (March, April, May), summer (June, July, August), fall (September, October, November), and winter (December, January, February), and the analyzed years were June 2015–May 2016 (“first”), June 2016–May 2017 (“second”), June 2017–May 2018 (“third”), and June 2018–May 2019 (“fourth”). We visualized the time course of all queries related to heartburn in the chosen countries, generating representative figures using the *ggplot2* and ggthemes2 packages of R 3.5.1 (R Foundation, Vienna, Austria) [25]. Trends were visualized using the geom_smooth() function from the ggplot2 package, which uses the Loess Regression method and 95% confidence intervals [25]. 

## 3. Results

The Keywords Planner generated 953 keywords for the English term “heartburn,” 1344 for the Polish term “zgaga,” and 1411 for the German term “Sodbrennen.” After evaluation of the list of generated keywords, 894, 1244, and 1388, respectively, of the terms were included in the final investigation. We presented the most popular keywords in each country in Appendix A. Results of descriptive analysis of the total number of searches and the categories are presented in Table 1 and Figure 1. The number of searches per 1000 Google-user years was as follows: 177.4 (AU), 178.1 (CA), 123.8 (DE), 199.7 (PL), 152.5 (UK), and 194.5 (US). The keyword categories with the highest number of queries were: “treatment,” “symptoms,” “diet,” and “causes of heartburn,” respectively.

### 3.1. Trends over Time

We visualized the number of heartburn-related searches conducted during the analyzed period for all chosen countries (Figure 2 and Figure 3). 

A comparison of the total number of queries in terms of years and seasons is presented in Table 2. and Table 3. For four countries (CA, DE, PL, and UK) we observed significant seasonal variability in the number of searches, with the lowest number occurring in summer (Table 2); there was no significant inter-season difference for AU and US. For each country, the relative differences between the seasons with the highest and lowest number of searches were as follows: 7% (AU), 12% (CA), 46% (DE), 41% (PL), 18% (UK) and 5% (US). In all regions except DE, there was a significant increase in heartburn-related queries over the analyzed period (Table 3). From the first year to the fourth year, the median number of searches per month increased by approximately 70% (AU), 18% (CA), 25% (DE), 27% (PL), 20% (UK), and 11% (US).

### 3.2. Heartburn-Related Keyword Categories

For all analyzed countries, we compared the number of searches relating to the heartburn-related keyword categories in terms of years and seasons (Appendix A). We consequently found significant seasonal variability in the number of searches associated with the keyword categories “treatment” (CA, DE, PL, UK, US), “home-based treatment” (DE, PL, UK), “herbal remedies” (AU, CA, DE, PL), “diet” (CA, DE, PL, UK), “what is heartburn?” (AU, CA, UK, US), “causes” (CA, DE, PL, UK), “symptoms” (CA, PL, US), and “pregnancy” (DE, PL). For countries from the northern hemisphere, the lowest number was mostly observed in summer, while in Australia the lowest number of searches was in winter or fall (Appendix A). Meanwhile, over the years examined there was a significant increase in the number of queries relating to “treatment” (AU, CA, PL, UK, US), “home-based treatment” (AU, CA, DE, UK, US), “herbal remedies” (DE, PL, UK), “diet” (AU, CA, DE, PL, UK, US), “what is heartburn?” (CA, DE, PL, US), “causes” (AU, CA, DE, US), “symptoms” (AU, CA, DE, PL, UK, US), and “heartburn in pregnancy” (AU, PL).

## 4. Discussion

In this infodemiology study, we found that the number of heartburn-related searches has increased over the last four years and may present a seasonal pattern in six Western countries. 

Infodemiology is a relatively new discipline of science, where web-based data are assessed for various determinants and distributions of information with an aim to inform the public and create tools useful for various policymakers [26]. Infodemiology has already been utilized to asses important health-related queries in the area of epidemiology of infectious diseases [27], mental health diseases [28], metabolic [29], and neurodegenerative diseases [30]. The most common sources of infodemiology data were Twitter, Wikipedia, and Google [31]. The global digital web-based footprints could serve as a powerful tool to expand the evidence-base medicine [32].

To the best of our knowledge, this is the first infodemiology study to use Google Ads to assess Internet users’ interest in heartburn-related information. Moreover, this is the first study to compare Google Ads data between different languages and different countries. 

We have chosen to use Google Ads Keyword Planner, which provides estimates of absolute search volume. In the context of heartburn, this tool can be used as a surrogate for the population-level burden of this condition [20,23]. This Google software enables access to monthly search volume data, which represents the total number of searches for selected keywords. To assess field related search volume, certain topic related words and phrases are entered into the Keyword Planner and the program finds keywords that are most relevant to the topic [21].

To control for any possible influence caused by the differing number of Google users in the analyzed countries, we expressed interest as the number of searches per 1000 Google-user years. Dent et al. estimated that the weekly incidence of GERD symptoms is five per 1000 person-years [5]; thus, our results suggest that one Google user with GERD symptoms generates 25–50 heartburn-related queries per year. Previously, Eusebi et al. reported that the prevalence of weekly GERD-related symptoms was approximately 10.0–14.9% in AU; 15.0–19.9% in CA, DE, and US; 20.0–24.9% in UK; and ≥25.0% in PL [4]. Indeed, we also found that, among these countries, Poland had the highest number of heartburn-related searches, while (among the countries that have English as a native language) the lowest number was from Australia. However, in our data, when all countries were considered, we found that interest in heartburn-related information was the lowest in DE, while Google users from CA and the UK showed similar levels of interest. This discrepancy between our findings and those of previous studies could be explained by the limited period of observation used in previous epidemiological studies. The results we obtained suggest that, if data are only collected during warm months, the prevalence may be underestimated, while if such collection is performed during cold months, overestimation may occur. Survey periods of less than one year and similarly short assessments of whether subjects meet GERD criteria limit the results of epidemiological studies [33,34]. Moreover, Eusebi et al. analyzed a few studies of GERD symptoms in Germany [35,36] and Poland [37].

Our data showed that Google users from DE are particularly interested in the treatment of heartburn. However, after examining the total number of queries related to heartburn treatment, we found that, for DE, the number of queries per 1000 Google user years was similar to that of PL [4]. As mentioned above Eusebi et al. found Poland to have the highest prevalence of GERD-related symptoms among the countries we analyzed. This may be attributable to the rich, fatty character of the local cuisine, which may explain the relatively high interest in PL in the associations between diet and heartburn.

We observed a seasonal variation in heartburn-related queries for CA, DE, PL, and UK; moreover, we observed a similar pattern for keyword categories: interest in heartburn-related information was lowest during warm months. Similarly, Hassid et al. reported modest seasonality in Google searches regarding diarrhea and vomiting in the US [38]. We also found that, for Polish-speaking Google users in Europe, interest in abdominal-pain-associated queries peaked in cold months (unpublished data). The observed variation could be related to the more prevalent outdoor activities during warm months. However, in the era of smartphones and constant access to the Web, this hypothesis requires further studies. The lack of a significant seasonal effect in AU in the US may be related to the distribution of the Google users in a large area which differs in climate. Many of the users might do not experience harsh winter or sharp changes in temperature between seasons. This could especially affect users living on the coast. A seasonal pattern of GERD was previously reported by Chen et al [39]. Specifically, in a study featuring over 76,000 participants from Taiwan, the authors found that, for both males and females, GERD incidence peaked in late autumn and early winter [39]. This phenomenon could be explained by seasonal variations in fat dietary intake; for instance, Shahar et al. reported that fat intake is higher in winter than in summer [40]. The presence of fatty foods in the duodenum can stimulate a release of cholecystokinin, which reduces the pressure of the lower esophageal sphincter, leading to gastroesophageal reflux [41]. Similarly, in winter, people generally increase their calorie intake and reduce their physical activity, which results in higher body mass index than in summer [42]; such a practice, through various mechanisms, may exacerbate GERD symptoms. Inversely, increased physical activity during spring may contribute to weight loss, which has been reported to improve GERD-related complaints [43]. Tobacco use, which is a risk factor of GERD [44], may also influence seasonal variability. Arku et al. reported that indoor nicotine levels may be significantly higher in winter than in summer [45]; similarly, Phillips reported that nonsmokers in Bremen, Germany, have higher exposure to environmental tobacco smoke in winter than in summer [46]. However, US tobacco sales are lowest in February and peak in June, which conflicts with this theory [47,48]. Finally, we hypothesize that internet e-commerce campaigns for heartburn-related drugs might be intensified during cold months. This may generate additional Internet traffic associated with heartburn. However, this hypothesis requires further investigation.

In all analyzed countries except DE, we observed, over the period examined, a significant growth in the number of queries associated with heartburn. In DE, we merely found a tendency in this regard. This may be caused by a high seasonal variation of searches in this country. Nevertheless, our results may reflect the growing prevalence of GERD in the Western world [3]. The highest relative growth in GERD interest was observed in Australia. In the US, there was a dramatic increase in the number of queries (by one million) from June 2015 to January 2016. The underlying reasons for this remain unknown; nevertheless, we can speculate that a surge in the consumption of fatty foods, an increase in the rate of Google users aged over forty years, a growing interest in self-management of heartburn, or the effects of possible intensified marketing campaigns for antacid drugs influenced this trend.

Olivera et al. previously reported that a substantial group of patients do not consult physicians about their heartburn [49]. For these individuals, consultation with “Dr. Google” (i.e., using Google to search for health-related information) may be an attractive alternative to visiting a health-care professional. In our data, Google users were particularly interested in the treatment of heartburn, which may reflect a high interest in self-management [50]. Since PPIs have now become available over the counter, health-related websites may recommend using this class of drugs; this may be a contributing factor to the statistic that approximately one in three patients with GERD use PPIs without consultation with a physician [51]. Such consumers are more likely to use PPIs sub-optimally and with inadequate symptom control [51]. Further, we identified a substantial number of queries regarding home-based or herbal remedies. Yoon et al. reported high use of dietary and herbal supplements among persons with GERD [52]. This may be driven by individuals for whom PPI treatment has failed due to frequent overlapping of refractory GERD with functional esophageal disorders [53,54], and who have consequently begun searching for another treatment method. The second most frequent were queries related to diet, which may be associated with self-education or an interest in identifying food products that trigger heartburn. The high number of searches asking “what is heartburn?” and those regarding heartburn symptoms may mirror the need for verification of unknown experienced sensations and/or unknown terms.

The simplicity of interacting with search engines and the comfort of avoiding the need to disclose embarrassing ailments may explain why “Dr. Google” is becoming a major consultant for Internet users [12]. For these reasons, the analysis of such queries might provide valuable insight into patients’ problems. The role of “Dr. Google” in searching for heartburn-related information is increasing, and this phenomenon should be noted by medical professionals as well as health-care regulators. In particular, there has been an observed increase in PPI consumption, which may be associated with heartburn self-management [55]. To reduce the overuse of PPIs and provide optimal treatment, and to avoid inadequate symptom control, Internet resources related to heartburn should encourage individuals to consult with a physician. Therefore, to provide reliable information it may be necessary for health-care professionals to both create and manage such a website [56].

Evaluating data sourced from Google Ads Keywords Planner has several limitations. First, we could only collect data pertaining to the last four years [20]. Second, the data do not provide the characteristics of the Google users, such as age, gender, place of residence (city or village), and detailed medical history; therefore, we could not conclude the possible etiology of heartburn. Previous studies have reported that women search for health-related information on the Internet more often than do men [57,58]; however, GERD is more prevalent in men [3]. Therefore, it is possible that the sex ratio among Google users interested in heartburn-related information is inverted. Third, the number of queries may only reflect the magnitude of the actual quantity of searches, not the prevalence of GERD. Google does not present the exact number of searches but only the ranges. Moreover, the company does not publish the methodology of the search volume calculations thus this limits confidence in results [20,21]. A substantial part of the total number of searches could be made by curious Google users who do not have heartburn. The search engine may automatically suggest the completion of search terms, which may bias users’ queries. Fourth, Google is used by ~88% of Internet users from the US, while in the other analyzed countries this rate exceeds 90% [16]. Thus, the total number of queries related to heartburn in the US might be underrepresented. To remove this discrepancy, we expressed the number of queries as a ratio per 1000 Google-user years. Fifth, we compared the number of queries across three different languages. The number of heartburn-related keywords differed between languages, which could contribute to the observed differences between the countries in the total number of searches per 1000 Google users, as well as in the number of searches associated with the specific categories. Moreover, symptoms but not diagnoses were searched and these might have their own wording and colloquial meanings in English spoken countries (USA, UK and Australia). Also, regional colloquialisms may exist in other non-English spoken and researched countries. Up to date none of the internet sources could offer adequate and reliable translations. Care and intuition should be thought before relying on machine translation of original sources [59].

## 5. Conclusions

In conclusion, interest in queries associated with heartburn has increased in the past four years in the countries analyzed and, in certain regions, a seasonal pattern may exist. The observed trends in heartburn-related searches may reflect the scale of the complaint, and should be verified through future epidemiological studies. These findings show that health-care providers should provide reliable information online to prevent patients from accessing information of unknown reliability.

## Figures and Tables

**Figure 1 ijerph-16-04591-f001:**
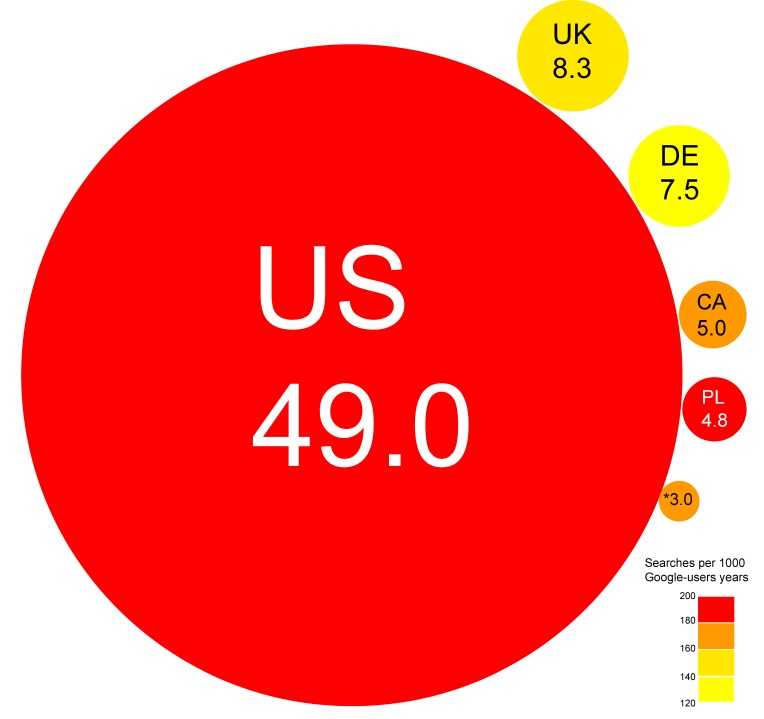
The number of heartburn-related queries per year in millions and the number of queries per 1000 Google-user years in each country. AU—Australia, CA—Canada, DE—Germany, PL—Poland, UK—the United Kingdom, US—the United States; rounds sizes are proportional to the number of heartburn-related queries per year. The color corresponds to the number of queries per 1000 Google-users years (darker color—higher number).

**Figure 2 ijerph-16-04591-f002:**
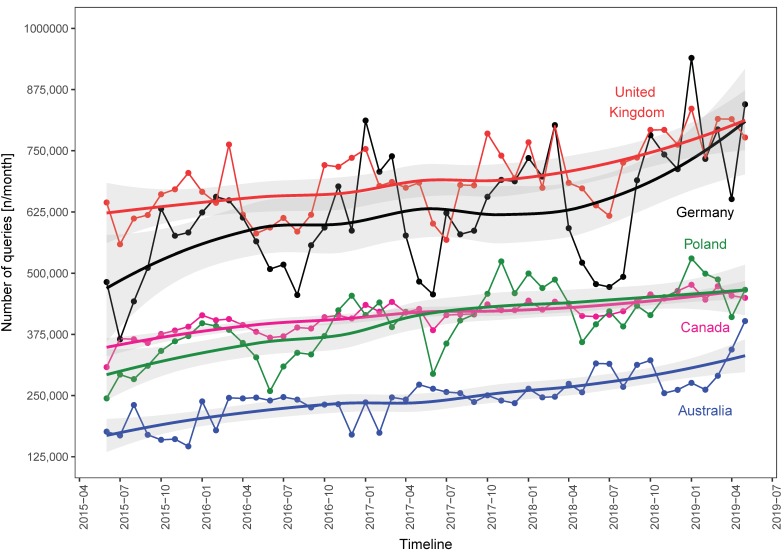
Number of heartburn-related Google queries from June 2015 to May 2019 in Australia, Canada, Germany, Poland, and The United Kingdom.

**Figure 3 ijerph-16-04591-f003:**
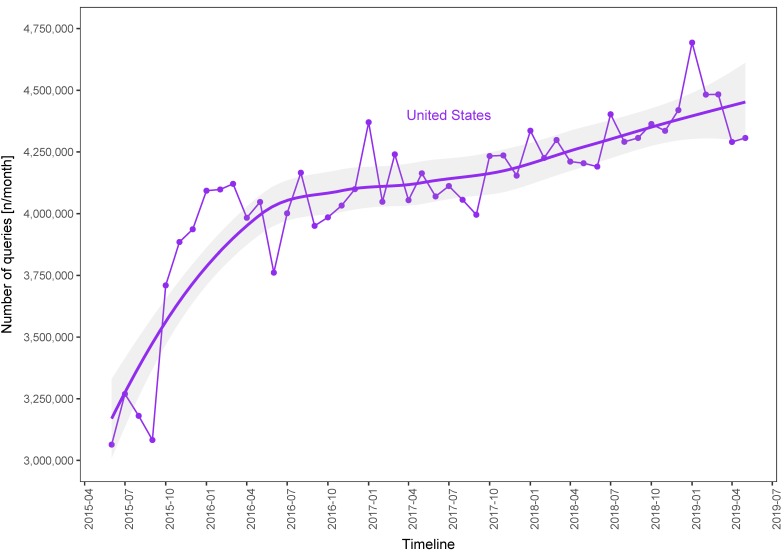
Number of heartburn-related Google queries from June 2015 to May 2019 in the United States.

**Table 1 ijerph-16-04591-t001:** Comparison between the analyzed countries regarding heartburn-related searches from June 2015 to May 2019.

Variables	Australia	Canada	Germany	Poland	The United Kingdom	The United States
Google users (mln)	16.6	27.9	60.3	24.0	54.4	252.0
Google search engine market share (%)	94.1	92.1	94.7	98.4	92.1	88.4
Total number of heartburn-related queries	11,777,020	19,878,045	29,869,865	19,170,430	33,194,490	196,037,375
	(177.4)	(178.1)	(123.8)	(199.7)	(152.5)	(194.5)
	(100.0%)	(100.0%)	(100.0%)	(100.0%)	(100.0%)	(100.0%)
Treatment	2,621,875	4,292,545	12,321,890	4,894,605	6,300,165	43,828,760
	(39.5)	(38.5)	(51.1)	(51.0)	(29.0)	(43.5)
	(22.3%)	(21.6%)	(41.3%)	(25.5%)	(19.0%)	(22.4%)
Home-based treatment	281,505	603,405	3,859,615	1,027,125	529,155	6,291,450
	(4.2)	(5.4)	(16.0)	(10.7)	(2.4)	(6.2)
	(2.4%)	(3.0%)	(12.9%)	(5.4%)	(1.6%)	(3.2%)
Herbal Remedies	222,230	456,140	309,125	227,425	670,915	5,121,765
	(3.3)	(4.1)	(1.3)	(2.4)	(3.1)	(5.1)
	(1.9%)	(2.3%)	(1.0%)	(1.2%)	(2.0%)	(2.6%)
Diet	565,995	1,326,825	3,020,700	2,043,985	1,724,810	13,340,085
	(8.5)	(11.9)	(12.5)	(21.3)	(7.9)	(13.2)
	(4.8%)	(6.7%)	(10.1%)	(10.7%)	(5.2%)	(6.8%)
What is heartburn?	681,335	878,275	420,160	308,780	1,387,865	9,278,745
	(10.3)	(7.9)	(1.7)	(3.2)	(6.4)	(9.2)
	(5.8%)	(4.4%)	(1.4%)	(1.6%)	(4.2%)	(4.7%)
Causes	946,825	1,011,130	1,170,705	727,500	3,310,060	8,323,060
	(14.3)	(9.1)	(4.9)	(7.6)	(15.2)	(8.3)
	(8.0%)	(5.1%)	(3.9%)	(3.8%)	(10.0%)	(4.2%)
Symptoms	1,543,550	1,555,925	768,170	1,624,020	3,079,495	14,090,405
	(23.2)	(13.9)	(3.2)	(16.9)	(14.2)	(14.0)
	(13.1%)	(7.8%)	(2.6%)	(8.5%)	(9.3%)	(7.2%)
Pregnancy	502,240	556,545	2,652,295	998,545	1,212,865	6,315,490
	(7.6)	(5.0)	(11.0)	(10.4)	(5.6)	(6.3)
	(4.3%)	(2.8%)	(8.9%)	(5.2%)	(3.7%)	(3.2%)

Data are presented as the total number of queries (number of queries: 1000 Google-user years) (percentage of total number).

**Table 2 ijerph-16-04591-t002:** Comparison regarding heartburn-related searches per month for each season.

Keyword Categories	Sp	Su	F	W	Differences between Seasons	Post-Hoc Test
All, AU	252,245 (245,823–278,236)	251,010 (237,648–264,998)	234,718 (211,893–251,624)	237,113 (177,743–261,780)	H(3) = 7.26; *p* = 0.06	-
All, CA	432,500 (411,221–443 861)	386,383 (367,905–414,604)	415,440 (386,088–438,069)	425,103 (412,505–444,550)	H(3) = 11.78; *p* < 0.01	Sp vs. Su: *p* = 0.009; Su vs. W: *p* = 0.009
All, DE	631,078 (573,800–752,476)	479,933 (456,481–510,810)	643,680 (584,064–689,920)	702,133 (647,904–733,763)	H(3) = 22.83; *p* < 0.001	Sp vs. Su: *p* < 0.001; Su vs. F: *p* < 0.001; Su vs. W: *p* < 0.001
All, PL	413,855 (378,118–445,215)	323,498 (290,604–392,360)	415,118 (356,075–437,983)	456,628 (410,374–477,133)	H(3) = 16.98; *p* < 0.001	Sp vs. Su: *p* = 0.02; Su vs. F: *p* = 0.02; Sp vs. W: *p* < 0.001
All, UK	685,568 (674,179–783,191)	612,100 (591,113–639,895)	718,958 (668,631–751,340)	719,670 (677,175–755,749)	H(3) = 16.10; *p* < 0.01	Sp vs. Su: *p* = 0.004; Sp vs. F: *p* < 0.001; Su vs. W: *p* = 0.002
All, US	4,207,538 (4,104,083–4,292,430)	4,062,750 (3,638,311–4,171,973)	4,013,993 (3,933,988–4,253,546)	4,189,768 (4,096,570–4,382,620)	H(3) = 6.37; *p* = 0.10	-

Note. AU: Australia, CA: Canada, DE: Germany, F: fall, PL: Poland, Sp: spring, Su: summer, UK: the United Kingdom, US: the United States, W: winter; Data are presented as medians (interquartile range).

**Table 3 ijerph-16-04591-t003:** Comparison of heartburn-related searches per month for each year.

Keywords Categories	1st Year	2nd Year	3rd Year	4th Year	Differences between Seasons	Post-Hoc Test
All, AU	177,658 (166,660–239,650)	238,030 (230,169–243,031)	252,768 (244,768–259,025)	301,708 (266,355–317,310)	H(3) = 32.03; *p* < 0.001	1st vs. 3rd: *p* < 0.001; 1st. 4th: *p* < 0.001 2nd vs. 3rd: *p* = 0.01; 2nd vs. 4th: *p* < 0.001 3rd vs. 4th: *p* < 0.001
All, CA	381,685 (366,026–396,723)	412,075 (388,623–422,866)	424,530 (415,586–436,820)	449,035 (437,770–458,530)	H(3) = 28.51; *p* < 0.001	1st vs. 2nd: *p* = 0.01; 1st vs. 3rd: *p* < 0.001; 1st vs. 4th: *p* < 0.001; 2nd vs. 4th: *p* < 0.001; 3rd vs. 4th: *p* = 0.01
All, DE	579,680 (503,728–625,795)	581,775 (515,340–684,631)	639,365 (584,765–692,158)	722,873 (611,633–784,663)	H(3) = 7.45; *p* = 0.06	-
All, PL	349,473 (306,336–374,835)	402,253 (336,690–421,814)	448,113 (392,473–474,101)	442,365 (413,446–471,501)	H(3) = 19.21; *p* < 0.001	1st vs. 3rd: *p* = 0.005; 1st vs. 4th: *p* < 0.001; 2nd vs. 4th: *p* = 0.03
All, UK	643,960 (616,890–667,300)	681,508 (617,634–718,104)	682,220 (673,995–746,873)	769,765 (732,391–798,230)	H(3) = 13.90; *p* < 0.01	1st vs. 4th: *p* < 0.009; 2nd vs. 4th: *p* < 0.009
All, US	3,910,660 (3,247,271–4,058,481)	4,051,168 (3,997,063–4,163,929)	4,207,538 (4,101,224–4,234,081)	4,349,405 (4,302,763–4,435,236)	H(3) = 30.76; *p* < 0.001	1st vs. 2nd: *p* = 0.02; 1st vs. 3rd: *p* < 0.001; 1st vs. 4th: *p* < 0.001; 2nd vs. 4th: *p* < 0.001; 3rd vs. 4th: *p* < 0.001;

Note. AU: Australia; CA: Canada; DE: Germany; F: fall; PL: Poland; Sp: spring; Su: summer; UK: the United Kingdom; US: the United States; W: winter. Data are presented as medians (interquartile range).

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
