# Peer review of "Heartburn-Related Internet Searches and Trends of Interest across Six Western Countries: A Four-Year Retrospective Analysis Using Google Ads Keyword Planner"

_ijerph, 2019, doi:10.3390/ijerph16234591_

Round 1

Reviewer 1 Report

Consulting Dr. Google about heartburn: Internet searches for and trends of interest in heartburn across six western countries

This paper investigates the frequency of ‘heartburn’ related internet searches across 6 different countries. Key findings suggest that the prevalence of such searches in rising globally and that a seasonality effect may exist, although not in all counties.

Infodemiology is a growing area that, here, seeks to exploit the internet to examine patterns that might reveal underlying attitudes and behaviours. For minor self-treatable ailments, in particular, such studies have the potential to provide revelations that might otherwise not be discernible. Google Ads Keyword planner provides a rich data set to be interrogated and contrasts with previous studies that rely on data sources such as Google Trends.

Overall this is an interesting study that I enjoyed reviewing. It demonstrates both how Google Ads can be exploited to reveal patient/consumer behaviour, and how search can be used as a proxy for incidence of heartburn across several western countries.

While grammatically correct, I suggest revising the title to: ‘Consulting Dr. Google about heartburn: Internet searches and trends of interest across six western countries’

‘Google Adwords Keywords Planner’ should be ‘Google Ads Keyword Planner’ or ‘Keyword Planner’ throughout as per https://ads.google.com/home/tools/keyword-planner/

Please define abbreviations on first use (i.e. GERD) particularly when they are not standard across all countries in the study.

Line 45. Consider changing ‘medical offices’ to the broader ‘health-care professionals’ here and elsewhere.

Line 47. Convenience is one factor. Other issues include privacy (less of an issue for GERD) and the reluctance to seek help for minor ailments (alluded to in line 54).

Line 77. For consistency, use ‘country’ rather than ‘region’. For example, ‘Scotland’ might be considered a region of the UK but no such regional analysis is conducted.

Line 78. It is unfortunate that the screenshot does not show the number of search queries. Where is this figure extracted? Similarly, it is unfortunate that the Google provided power-of-10 ranges are literally an order of magnitude apart.

Line 98. ‘research’ should be replaced with ‘search’.

Section 2.2 (Line 98-102) should be rewritten for clarity:

The total number is the total of searches for all retained related queries across all 48 months. Is the ratio over ALL Google searches presented? I expect this value to be very low. Can you provide (in the appendix) a table showing related English search terms (say the top 20) and their respective search counts for a representative month?

Line 119. A large number of keywords are retained in the final investigation. While there are many variants of possible search terms, you should provide reassurance that your inclusion criteria are valid. One imagines that Zipf's law would apply, with keywords searches becoming increasing obscure. For robustness, do similar results hold if only (say) the top 10 related searches are included? The US pattern looks very different based on a single ‘heartburn’ term search in Google Trends.

Table 1. It would be helpful to format so the data consistently appears over either 2 or 3 lines.

Figure 1. The label for Australia [AU] is missing. The caption should indicate that the numbers are in millions.

Line 150. Statistically speaking, DE does not exhibit a significant increase in queries. It would be worth commenting further given the apparent DE trend in Figure 1.

Line 178. Inform public [policy|debate]?

Line 227-32. The prevalence of indoor (and online) activities during dark winter months cannot be discounted. Can you be sure that increased search activity does not simply reflect increased time to dwell on health-related issues? Similarly, while the Christmas period may coincide with excess consumption (and related symptoms), new year may also be a time to reflect and address previously ignored maladies. Interestingly Google Trends appears to indicate the northern US states are associated with higher search prevalence compared with southern states. The lack of a significant seasonal effect may simply reflect the vastness of the US and Australia as continents, and large (sub) populations who do not experience a harsh winter.

Line 238. This is an interesting hypothesis. As indicated in the supplementary file screenshot, could the price for paid google ads (search ranking) indicate advertising demand?

Can you comment on the leading remedies (i.e. over the counter products) for heartburn treatment? Can you present any evidence to suggest sales of these products have followed the same seasonal or year-on-year trends?

Line 245-49. Growth in search does not necessarily mean growth in prevalence. As you acknowledge, growing absolute search counts may reflect growing sub-populations who use internet searches to investigate health-related issues (for example, through rising ‘computer literacy’ in older generations). Can you present estimates from other studies that might help to decompose these confounding factors? Perhaps you could use a baseline condition for comparison that is known to have been relatively static in prevalence.

Line 271-276. Given Google’s dominance (Line 289), and the ability to pay for ads that appear at the top of search results, it would be worth commenting on practical considerations for policymakers and healthcare professionals. For example, what is the point of creating a website if searchers are not directed to it? What ethical responsibility do search and social media platforms have to ensure appropriate content is presented? Can the industry be trusted to self-regulate? Could targeted ads be used for public health campaigns (see Serrano, 2016)?

Line  285. Data presented in table 1 suggests search counts are rounded to the nearest 5?

References

Serrano, W.C., Chren, M.M., Resneck, J.S., Aji, N.N., Pagoto, S. and Linos, E., 2016. Online advertising for cancer prevention: Google ads and tanning beds. JAMA dermatology, 152(1), pp.101-102.

Author Response

Dear Reviewer,

thank you for the very fast review and the very detailed comment. We are pleased that you enjoyed our manuscript. We discussed our comments and revised the manuscript. We hope that the new version meets your expectations. For our convenience, most of the revised sentences are included in replies to our comments. All changes are highlighted.

While grammatically correct, I suggest revising the title to: ‘Consulting Dr. Google about heartburn: Internet searches and trends of interest across six western countries’

Dear Reviewer,
we propose a new title

„Heartburn-related Internet searches and trends of interest across six western countries: a four-year retrospective analysis using Google Ads Keyword Planner”

Google Adwords Keywords Planner’ should be ‘Google Ads Keyword Planner’ or ‘Keyword Planner’ throughout as per https://ads.google.com/home/tools/keyword-planner/

Thank you for these comments.
Previously, the official name was Google Adwords Keywords Planner, similarly to the title of the cited article:

20. Zink, A.; Schuster, B.; Rüth, M.; Pereira, M.P.; Philipp-Dormston, W.G.; Biedermann, T.; Ständer, S. Medical needs and major complaints related to pruritus in Germany: a 4-year retrospective analysis using Google AdWords Keyword Planner. Journal of the European Academy of Dermatology and Venereology 2019, 33, 151–156.

However, the change of the official name requires from us to update the name of keywords generator. In the revised manuscript you may find term „Google Ads Keyword Planner”.

Please define abbreviations on first use (i.e. GERD) particularly when they are not standard across all countries in the study.

GERD abbreviation has already been defined at the beginning of the Introduction. All first use abbreviations have been checked and expanded.

Line 45. Consider changing ‘medical offices’ to the broader ‘health-care professionals’ here and elsewhere.

In the revised manuscript we changed term 'medical offices' to the proposed 'health-care professionals'

Line 47. Convenience is one factor. Other issues include privacy (less of an issue for GERD) and the reluctance to seek help for minor ailments (alluded to in line 54).

New version of the sentences:

1)

Further, the Internet provides the convenience of immediate access to resources associated with the condition in question, as well as the comfort of potential support from other users and anonymity [12].

2)

For instance, people may experience moderate ailment and search for relief on the Internet which may be explained by a reluctance to seek help for minor ailments in health-care professional office.

Line 77. For consistency, use ‘country’ rather than ‘region’. For example, ‘Scotland’ might be considered a region of the UK but no such regional analysis is conducted.

For better clarity, we added a sentence:

The considered regions were countries.

Line 78. It is unfortunate that the screenshot does not show the number of search queries. Where is this figure extracted? Similarly, it is unfortunate that the Google provided power-of-10 ranges are literally an order of magnitude apart.

The Google Ads Keyword Planner shows the only range of search volume for each keyword. This problem is discussed further in the same paragraph as well as in limitations.

Line 98. ‘research’ should be replaced with ‘search’.

Corrected.

Section 2.2 (Line 98-102) should be rewritten for clarity:

The total number is the total of searches for all retained related queries across all 48 months. Is the ratio over ALL Google searches presented? I expect this value to be very low. Can you provide (in the appendix) a table showing related English search terms (say the top 20) and their respective search counts for a representative month?

As you may notice in Figure S1 the number of searches of keyword „heartburn” in one month in the UK ranges between 10K-100K (mean 55K). Beside of this keyword are 894 keywords which were included.

We rewritten the lines 98-102:

For all countries, we calculated the arithmetical mean of the sum of monthly search volume for each keyword category and all keywords. The data was presented as the total number of queries during the analyzed period for each keyword category and all keywords. Additionally, we divided the total search volume of each category by the total number of searches in the analyzed period, and separately by the number of Google users in each country, as declared by the Keywords Planner. Additionally, we presented mean Google search engine market share in investigated countries in the years 2015-2019 [16].

Moreover, we added a new table to supplementary (new Table S1) that presents the top 20 search terms for each analyzed country. We hope that this will answer your interest.

Line 119. A large number of keywords are retained in the final investigation. While there are many variants of possible search terms, you should provide reassurance that your inclusion criteria are valid. One imagines that Zipf's law would apply, with keywords searches becoming increasing obscure. For robustness, do similar results hold if only (say) the top 10 related searches are included?

We added the following sentence to material and methods:

Two of three authors (M.K., I.Ł., A.M.) independently analyzed the generated list of keywords and create the final lists. Any inconsistencies in the lists were referred by the senior author (W.M.).

As mentioned above we included top 20 related searches in Table S1.

The US pattern looks very different based on a single ‘heartburn’ term search in Google Trends.

We do not agree with your observation. The US pattern reflects the Google Trends course over time:

During the first year in both Google Trends and Google Ads the search volume raised by 25% (from 3 million to 4 million, from 55-60 RSV to 75 RSV. Further, the Google Trends peaks correspond with the peaks in Figure 3. We suspect that this is an observatory error due to different proportions of the figure from Google Trends and Figure 3.

Table 1. It would be helpful to format so the data consistently appears over either 2 or 3 lines.

We corrected the table and results appear over 3 lines in all cells.

Figure 1. The label for Australia [AU] is missing. The caption should indicate that the numbers are in millions.

The caption was rewritten:

Figure 1. Number of heartburn-related queries per year in millions and number of queries per 1000 Google-user years in each country.

In the new Figure 1 we included label AU for Australia and rewritten the legend.

Line 150. Statistically speaking, DE does not exhibit a significant increase in queries. It would be worth commenting further given the apparent DE trend in Figure 1.

We added one sentence to the discussion:

„In DE, we merely found a tendency in this regard. This may be caused by a high seasonal variation of searches in this country.

Line 178. Inform public [policy|debate]?

We meant public as 'society', 'community'.

Line 227-32. The prevalence of indoor (and online) activities during dark winter months cannot be discounted. Can you be sure that increased search activity does not simply reflect increased time to dwell on health-related issues?

We added a following sentence:

We observed a seasonal variation in heartburn-related queries for CA, DE, PL, and UK; moreover, we observed a similar pattern for keyword categories: interest in heartburn-related information was lowest during warm months. Similarly, Hassid et al. reported modest seasonality in Google searches regarding diarrhea and vomiting in US [38] .We also found that, for Polish-speaking Google users in Europe, interest in abdominal-pain-associated queries peaked in cold months (unpublished data). The observed variation could be related with the more prevelent outdoor activities during warm monts. However, in the era of smartphones and constant access to the Web this hypothesis requres further studies.

Similarly, while the Christmas period may coincide with excess consumption (and related symptoms), new year may also be a time to reflect and address previously ignored maladies.

This is an interesting point but we focused mostly on the comparison between each season. Figures 2 & 3 presents some examples: lower interest in December than in January. However, it may be related to the lower Internet use during the Christmas celebration in comparison with the next month.

Interestingly Google Trends appears to indicate the northern US states are associated with higher search prevalence compared with southern states. The lack of a significant seasonal effect may simply reflect the vastness of the US and Australia as continents, and large (sub) populations who do not experience a harsh winter.

We added this excellent observation to the discussion:

The observed variation could be related to the more prevalent outdoor activities during warm months. However, in the era of smartphones and constant access to the Web, this hypothesis requires further studies. The lack of a significant seasonal effect in AU in US may be related to the distribution of the Google users in a large area which differs in climate. Many of the users might do not experience harsh winter or sharp changes in temperature between seasons. This could especially affect users living on the coast.

Line 238. This is an interesting hypothesis. As indicated in the supplementary file screenshot, could the price for paid google ads (search ranking) indicate advertising demand?

This is a technical aspect. The default order correlates with search volume and advertising demand but does not fully reflect it.

Can you comment on the leading remedies (i.e. over the counter products) for heartburn treatment? Can you present any evidence to suggest sales of these products have followed the same seasonal or year-on-year trends?

To our best knowledge, no open-source data on the trend of PPI sales for the analyzed periods are available. Therefore we did not confront our results with real-world marketing trends. We only found an article from Iceland (https://www.ncbi.nlm.nih.gov/pmc/articles/PMC5977421/) presenting that after 13 years the overall consumption of PPIs increases.

Line 245-49. Growth in search does not necessarily mean growth in prevalence. As you acknowledge, growing absolute search counts may reflect growing sub-populations who use internet searches to investigate health-related issues (for example, through rising ‘computer literacy’ in older generations). Can you present estimates from other studies that might help to decompose these confounding factors? Perhaps you could use a baseline condition for comparison that is known to have been relatively static in prevalence.

The Google Ads Keyword Planner does not adjust the search volume to the current number of Google users as Google Trends. This is the limitation of the method and is further discussed, Nevertheless, the Planner provides an estimation of search volume per 1000 Google-users as well as the area of interest of the Google users.
A combination of Keyword Planner and Google Trends may be useful. However, we did not find any study combining both methods.

Line 271-276. Given Google’s dominance (Line 289), and the ability to pay for ads that appear at the top of search results, it would be worth commenting on practical considerations for policymakers and healthcare professionals. For example, what is the point of creating a website if searchers are not directed to it?

Your comment could be an interesting background for further study or at least a letter to the Editor. The positioning of the websites recommended by patients federations and scientific societies in the search engine is a technical problem that could be faced by using professional e-marketing service (SEO). This could be a slow process but the organic growth is recognized but Google algorithms recognize this 'natural' expansion and promote such websites. (oral communication from specialists of e-marketing and knowledge from e-course from Google https://learndigital.withgoogle.com/digitalgarage/course/digital-marketing)

What ethical responsibility do search and social media platforms have to ensure appropriate content is presented? Can the industry be trusted to self-regulate?

The main ethical responsibility of search engines and social media is to provide an environment for free expression and free discussion. In our opinion, the critical is the responsibility of the authors of such content. Therefore, we should disseminate the good culture in e-discourse. Firstly, all e-content should be signed by own name of the author. Secondly, the authors should disclose financial relations with the industry.

The popular marketing slogan states: „Content is a king”. We believe that creating e-sources characterized by high quality with the recommendation of scientific and patients authorities will win the competition in search engines and media platforms with unreliable content.

We do not support the call for regulation of e-media giants due to 1) intervention in private initiative 2) hard implementation of such regulations.

Could targeted ads be used for public health campaigns (see Serrano, 2016)?

References

Serrano, W.C., Chren, M.M., Resneck, J.S., Aji, N.N., Pagoto, S. and Linos, E., 2016. Online advertising for cancer prevention: Google ads and tanning beds. JAMA dermatology, 152(1), pp.101-102.

This is a broad topic and is beyond the scope of our study. We aimed to present the utility of Google Ads for epidemiological purposes. Google Ads Keyword Planner may indicate the most accurate keywords for targeted campaigns. It will be interesting how much money it requires to spend to convince one Google user to come to a medical facility to e.g. oncological screening.

Line  285. Data presented in table 1 suggests search counts are rounded to the nearest 5?

The lowest range of search volume of keyword was 0-10 thus the lowest mean search volume per month was 5.

Reviewer 2 Report

The paper is interesting, new, clear and may suggest further studies.

Just some observation:

line 125 a possible improvement to make more comparable the data could have been to see also at the ratio between searches and total population of each country, but this is not essential for publication line 139 the Figure 2 is only partially visible in my PDF copy line 279 gender difference could be a strongly significant factor, but Google does not provide this, as you wrote!

Author Response

Dear Reviewer,

thank you for the fast review and your comments. We discussed our comments and revised the manuscript. We hope that the new version meets your expectations. For our convenience, most of the revised sentences are included in replies to our comments. All changes are highlighted.

line 125 a possible improvement to make more comparable the data could have been to see also at the ratio between searches and total population of each country, but this is not essential for publication 

Dear Reviewer,

we resigned from the comparison of the ratio between searches and the total population of each country due to the lower popularity of Google search engine among users from the United States (this problem is also discussed in limitations). For this reason, we decided to present the ratio between search and the total population of Google users in each country.

line 139 the Figure 2 is only partially visible in my PDF copy

Dear Reviewer,

we checked both .pdf and .docx versions of the manuscript and Figure 2 was fully visible. Nevertheless, we will carefully check the quality and presentation of the figures in the next versions of the manuscript.

line 279 gender difference could be a strongly significant factor, but Google does not provide this, as you wrote! 

We exchange the word „sex” for „gender” inline 279.

Thank you for all comments and fast reviews.